# Living with Gluten and Other Food Intolerances: Self-Reported Diagnoses and Management

**DOI:** 10.3390/nu12061892

**Published:** 2020-06-26

**Authors:** Magdalena Araya, Karla A. Bascuñán, Dana Alarcón-Sajarópulos, Francisco Cabrera-Chávez, Amaya Oyarzún, Alan Fernández, Noé Ontiveros

**Affiliations:** 1Institute of Nutrition and Food Technology (INTA), University of Chile, Santiago 7830490, Chile; karlabascunan@gmail.com (K.A.B.); dana.alarcon@hotmail.com (D.A.-S.); amaya.oyarzun@gmail.com (A.O.); 2Department of Nutrition, Faculty of Medicine, University of Chile, Santiago 8380453, Chile; alfernandezbe@gmail.com; 3Faculty of Nutrition and Gastronomy Sciences, Master of Science Graduate Program in Nutrition and Medicinal Foods, University of Sinaloa, Culiacán, Sinaloa 80019, Mexico; fcabrera@uas.edu.mx; 4Division of Sciences and Engineering, Department of Chemical, Biological and Agricultural Sciences, Clinical and Research Laboratory (LACIUS, URS), University of Sonora, Navojoa, Sonora 85880, Mexico

**Keywords:** food intolerance, celiac disease, gluten sensitivity, gluten-free diet

## Abstract

People suffering from a food intolerance (FI) tend to initiate restrictive diets such as a gluten-free diet (GFD), to alleviate their symptoms. To learn about how people live with these problems in daily life (independent of their medical diagnoses), 1203 participants answered a previously validated questionnaire and were divided into: G1 (those self-reporting symptoms after gluten consumption) and G2 (those informing no discomfort after gluten consumption). Self-reported clinical characteristics, diagnoses and diets followed were registered. Twenty nine percent referred some FI (8.5% in G1). In G1, self-reported diagnoses were more frequent (*p* < 0.0001), including a high proportion of eating and mood disorders. Diagnoses were reported to be given by a physician, but GFD was indicated by professional and nonprofessional persons. In G2, despite declaring no symptoms after gluten consumption, 11.1% followed a GFD. The most frequent answer in both groups was that GFD was followed “to care for my health”, suggesting that some celiac patients do not acknowledge it as treatment. Conclusion: close to one third of the population report suffering from some FI. Those perceiving themselves as gluten intolerant report more diseases (*p* < 0.0001). A GFD is followed by ~11% of those declaring no symptoms after gluten ingestion. This diet is perceived as a healthy eating option.

## 1. Introduction

Eating is one of the primary and fundamental activities that human beings do to remain alive. During the last century, eating habits in the population have changed drastically and this coincides with relevant changes in the patterns of disease being observed [1,2]. Changes are multifactorial, some being related to what and in what quantity people choose to eat and how foods are processed. Food intolerance (FI) is defined as an adverse reaction to food with or without immune participation. It has broad gastrointestinal and extraintestinal clinical manifestations and its frequency seems to be increasing [2,3,4], in occasions being described in up to 20% of the population [3,5]. Persons suffering from FI tend to restrict their diets in an effort to improve their symptoms [6,7]. A gluten-free diet (GFD) is one of the diets patients currently choose, following their perceptions and often without being diagnosed with gluten-related conditions [8,9]. Although without scientific support, at present this diet represents an eating fashion/trend for many who consider it healthier, useful for losing weight and even improving performance in sports [10,11].

Although a GFD represents the treatment for celiac disease (CD), non-celiac gluten/wheat sensitivity (NCG/WS) and wheat allergy (WA) [12], consuming gluten does not result in damage to health for those who follow it as a lifestyle or fashion. This latter group has increased so much that it has made a considerable contribution to the current increase of the gluten-free products market. Thus, today the food market is full of so-called “no” products: free of gluten, casein, soy, lactose and/or other substances, all of which make it really difficult for patients differentiating between products that are suitable for the treatment of gluten-related diseases and those that are simply options for the current fashion/trends. We consider it of interest to learn how people live with these kinds of symptoms and diets in daily life and, with this in mind, set as the objective of this study to characterize self-reported intolerance to gluten and other foods in Chilean adults, in a real-life context.

## 2. Materials and Methods

Individuals 18–65 years of age, contacted in places close to malls, parks and metro entrances in areas where middle-income families live in Santiago, the capital of Chile, from July to October 2019, were assessed. They were invited to participate in a short interview conducted by a person trained to fill a previously validated questionnaire [13]. Those who accepted signed an informed consent and general information about sex, age and schooling was then registered. Participants that answered the first question declaring complaints/symptoms after eating gluten-containing foods, formed group 1 (G1). Those who referred no discomfort after gluten consumption formed group 2 (G2). The questionnaire then went on to collect data about the frequency of food intolerances (gluten-related and others), associated symptoms, the person/health professional who made the diagnosis and/or was asked for advice, and what kind of diets people followed in order to manage their symptoms. Answers represent self-reported symptoms and self-reported diagnoses, meaning what people feel and think they suffer and live with accordingly, but do not provide information about diagnoses of certainty. Based on the answers obtained, conditions were grouped into gastrointestinal (irritable bowel syndrome, lactose intolerance, chronic diarrhea and gastrointestinal cancer) and extraintestinal (non-wheat/gluten allergies, autoimmune diseases, eating disorder and mood disorder). In G2, special emphasis was given to why they followed the special diets and how strict these were.

Statistical analysis included describing data as mean ± standard deviation (SD), assessment by graphical inspection and the Shapiro–Wilk test, Student’s *t*-test for comparison of background independent variables and Chi-squared or Fisher’s exact test for comparison of gender and education years distribution between groups. The same tests were used to compare presence of self-reported symptoms and self-reported diagnoses as well as GFD distribution between groups. A 5% difference defined the significance level and the software packages STATA^®^ v. 13.1 (StataCorp LLC, College Station, TX, USA) and GraphPad Prism v. 6 (GraphPad Software, La Jolla, CA, USA) were used for analysis and figures processing.

## 3. Results

Of the 1203 persons that answered the questionnaire, 349 (29%) referred to some kind of food intolerance and 102 (8.5%) of them reported symptoms after gluten intake (G1). Of 1101 declaring no symptoms after eating gluten (G2), 247 (20.5%) reported developing other/undefined food intolerances (Figure 1). General characteristics in G1 and G2 are shown in Table 1. Participants belonging to G1 were older and were a higher proportion of women. A total of 151 self-reported diagnoses were declared in G1 (Table 2), 88 and 63 gastrointestinal and extraintestinal conditions, respectively; In G2, 668 diagnoses were reported, of which 399 and 269 were gastrointestinal and extraintestinal, respectively.

### 3.1. Group 1 (G1)

75/102 individuals (73.5%) stated that they knew they suffered from at least one medical condition, but symptoms remained undiagnosed in 26.5% of them (Figure 1).

Table 2 describes the gastrointestinal and extraintestinal self-reported diagnoses and Table 3 those that were reported to diagnose and prescribe GFD. Of participants declaring to follow a GFD, only one knew that additives and cross-contamination are factors that must be considered when maintaining strict GFD.

### 3.2. Group 2 (G2)

Food intolerance, meaning developing any clinical manifestation after eating foods not containing gluten/wheat, was reported by 247/1101 cases (22.4%) (Figure 2). Self-reported conditions are shown in Table 2: 86.1% of them were non-gluten related and 86.1% gluten-related symptoms/diagnoses, while 14.7% complaints referred to neurological manifestations of gluten sensitivity.

When asked whether they followed some special diet to alleviate their symptoms, despite their initial declaration of not experiencing discomfort after gluten consumption, 122/1101 (11.1%) declared to follow a GFD; justifications for this diet were: “controlling weight” (*n* = 48), “it is healthier” (*n* = 71), “I have a relative with CD” (*n* = 2) and that gluten-free foods “taste better” (*n* = 1).

### 3.3. Gluten-Free Diet

Of the 1203 persons interviewed, 16.4% followed a GFD (Figure 2), 74.5% of those belonged to G1 and 11% to G2. In both, a GFD was prescribed/advised by professionals and nonprofessionals, without differences between groups (NS, Table 3). The GFD was followed “strictly”/“do my best to make it strict” was the most frequent answer in both groups (NS). When requested to explain why they followed it, as expected in G1 the main reason was indeed the appearance of symptoms. However, 16/102 (26.7%) in this group mentioned that they followed the diet because it was healthier, as if they were not aware that this was their treatment. In G1 and G2, 23% and 77% of participants, respectively, declared that they follow a GFD to “care for my health”.

## 4. Discussion

How many people follow a GFD is a question that remains unclear. In this study, using a real-life approach, we show that close to one third (29%) of the adult population interviewed declared to suffer from FI and modified their diet to alleviate their symptoms. Although the available evidence shows that this high percentage of positive cases decreases to 1–2% when diagnosed by a double-blind placebo-controlled food challenge [14], these results are relevant to understanding what people think about their health, what they believe and how they modify their daily life accordingly. In G1, individuals stated that they suffered significantly more gastrointestinal and extraintestinal conditions compared with G2 (*p* = 0.0001). Extraintestinal diagnoses related to gluten/wheat consumption were more than twice (44.1%) the ones reported in G2 (21.3%). It was unexpected that of the 151 diagnoses declared in G1, 24.7% referred to neurological manifestations (*p* < 0001). These kind of manifestations of gluten sensitivity, like ataxia [15], epilepsy [16], peripheral neuropathy [17] or myopathy [18] have been described for many years [19], but it is rather recently that symptoms like migraine [20] and anxiety/depression [21] are described in association with CD and NCGS [22]. Autoimmune, genetic, epigenetic, microbiome and other interactions play relevant roles in CD pathophysiology, but the mechanisms explaining the relationship between CD and other conditions characterized by gluten sensitivity—like neurological manifestations of gluten sensitivity—are still incompletely understood [23]. Due to the methodology applied in this study, the results could not be analyzed any further and clearly show the need for further investigation of the intestine-brain axis.

In this series, food intolerance appears more frequently (29%) than reported by Lomer (who estimated it at 15–20% [5]) and Choung (1.7%) [24]. Studies of food intolerances at present refer mainly to irritable bowel syndrome [25] and more recently to the effects of diets reduced in FODMAPs (fermentable, oligosaccharides, disaccharides, monosaccharides, and polyols) [19,20]. It is of interest to discuss the potential role of FODMAPS as symptom inducers. A low FODMAP diet has been described to improve gastrointestinal symptoms in patients with IBS [26]. Additionally, in patients with CD that report persisting gastrointestinal symptoms while following a GFD, addition of a short-term, low-FODMAP diet helped improve the symptoms and their psychological health [27]. Given the current extensive range of complaints associated with gluten consumption, it is not surprising that people intuitively restrict their diets in order to improve symptoms they do not understand, but are clearly related to the act of eating. Unfortunately, these aspects were not included in the questionnaire we used and limit the data analysis. In recent years, the number of subjects reported being on a GFD has greatly increased, ranging from 6.2 to 13% in the general population [28]. This is partly due to the perception that a GFD is a healthier dietary option in comparison to a full diet, even for people that are not affected by CD, NCW/GS or WA [9]. However at the same time, in the last decades there is an increase of autoimmune conditions in the population [29] and celiac patients develop several of them more frequently than the general population [30]. That a GFD is the treatment of CD is routinely explained to recently diagnosed patients so, that most persons in G1 were unaware of the risks of not following proper dietary treatment raises concern and emphasizes the need to correctly separate those at risk of autoimmunity from those that simply follow an eating fashion/trend.

It is interesting that participants in G2 were assigned to this group—specifically—because they reported no discomfort after eating gluten-containing foods in the initial question, nevertheless, 11% subsequently declared that they follow a GFD. This is relevant because to date, the available evidence does not support health benefits for a GFD in the absence of gluten-related disorders [9,31]. Gluten-free foods often have lower nutritional value (less protein, more glucose and saturated fats, less fiber) [10]. In addition, a higher content of blood heavy metals levels has been described in persons following a GFD [11]. Furthermore, a GFD usually has a negative economic impact due to the higher prices of gluten-free products, which is especially burdensome for lower income families having a celiac member. In a previous study in Chile, Estevez [32] assessed the gluten-free family basket, reporting that this was 42% less available than the regular family basket, three times more expensive (up to 500% higher for breads), with up to a 69% lower protein content and with no fortifications.

The only response to the question “who gave the diagnosis” to the participant was “a physician”, suggesting that the whole group interviewed had access to medical care. However, diets were not necessarily prescribed by the physician diagnosing the patient. Instead, participants reported even healers and self-administered diets, strongly suggesting that the time assigned to the patient’s physician appointment may be insufficient in public health systems. This result differs from those reported in previous studies conducted in Hispanic America (Mexico, Colombia, Argentine, El Salvador and Brazil), where the same validated questionnaire was applied but used different data analysis, and where individuals were diagnosed by a variety of persons [13,33,34,35,36]. The fact that nearly all participants in this study declared that they follow a strict GFD disagrees with many studies that evaluate adherence, including our own previous studies [37]. Difficulties measuring adherence to diets are well-known [38]. Adherence to a GFD was not the objective of this study and, given the questionnaire, applied evaluation of these results was limited.

Finally, why gluten seems to be increasingly inducing symptoms in the population is an interesting matter of debate. It is only in recent years that efforts have been made to understand whether and why specific ingredients or food components, like gluten, may trigger symptoms [5]. Two arguments in this debate deserve attention: today, the food industry uses gluten very frequently and in large amounts as a food component or additive and this has led to a three or more times increase of gluten consumption in the population in the last 50 years [39]. Furthermore, the more drastic techniques applied during modern food processing may change the molecular configuration of processed-food components therefore potentially modifying immune recognition phenomena [40]. How these factors may relate to symptom appearance or to the larger variety of symptoms observed in the different body systems (CNS, skin and autoimmune diseases among others) is an interesting area for future research.

## 5. Conclusions

Results show that FI as perceived by apparently healthy adults is increasing—nearly one third of the apparently healthy population reports to suffer from it. Those that associate their problems to gluten consumption report significantly more gastrointestinal and extraintestinal conditions than those that report other FIs, neurological manifestations representing nearly 25% of the reports. A GFD is considered a healthy alternative to manage unexplained symptoms, even when there is no diagnosis justifying a restrictive diet. The health system may be efficient in making a diagnosis, but seems unable to treat, educate and follow the affected people effectively. These results reveal interesting areas that deserve further research.

## Figures and Tables

**Figure 1 nutrients-12-01892-f001:**
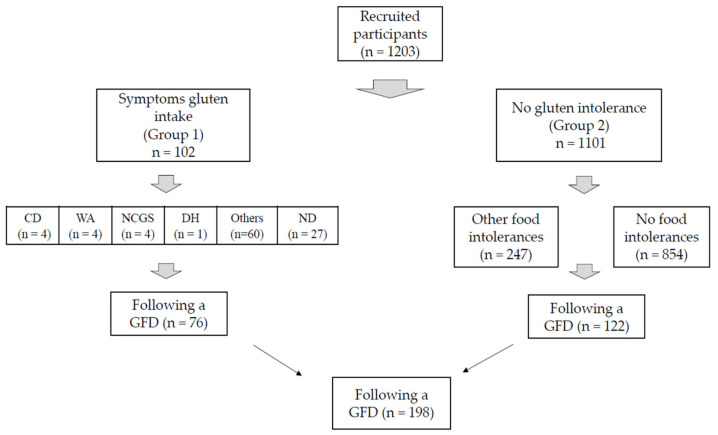
Flowchart and description of self-reported food intolerances in the study groups. Celiac disease, CD; wheat allergy, WA; non-celiac gluten sensitivity, NCGS; dermatitis herpetiformis, DH; no diagnosis, ND; gluten free diet, GFD.

**Figure 2 nutrients-12-01892-f002:**
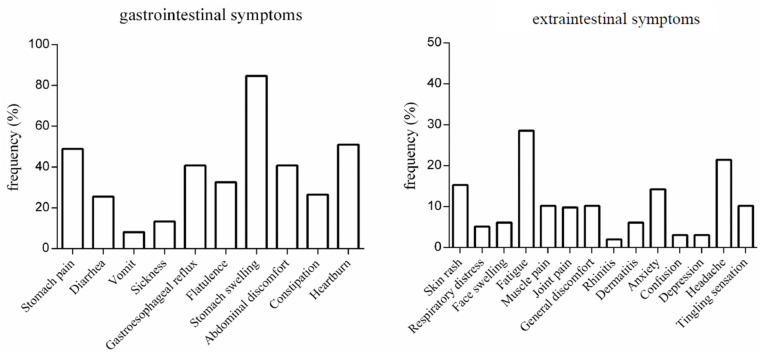
Self-reported gastrointestinal and extraintestinal symptoms associated with gluten consumption.

**Table 1 nutrients-12-01892-t001:** General characteristics and conditions self-reported by 1203 apparently healthy adults.

Variables	Group 1 (*n* = 102)	Group 2 (*n* = 1101)	*p*-Value *
Age, years	39.1 ± 17.5	35.2 ± 16.1	0.033
Female, *n* (%)	78 (76.5)	528 (48)	0.0001
Education			0.124
<8 years, *n* (%)	10 (9.8)	116 (10.3)	
≥8–<12 years, *n* (%)	44 (43.1)	570 (51.8)	
≥12 years, *n* (%)	48 (47.1)	401 (36.4)	
*Diagnoses reported*			
Gastrointestinal diagnosis ^†,^**, *n* (%)	75 (73.5)	320 (29.0)	0.0001
Extraintestinal diagnosis ^††^, *n* (%)	45 (44.1)	235 (21.3)	0.0001
Gluten-free diet, *n* (%)	76 (74.5)	122 (11.0)	0.001

Data are shown as frequency (percentage) except for age (mean ± SD). * *p*-value obtained by comparison of groups using the independent *t*-test. ** one person reported a gastrointestinal and an extraintestinal diagnosis of gastrointestinal cancer. ^†^ defined as self-reports of one or more gastrointestinal conditions/symptoms (including irritable bowel syndrome, lactose intolerance, chronic diarrhea and gastrointestinal cancer). ^††^ defined as reporting one or more extraintestinal symptoms, including allergies, autoimmune diseases, eating disorder and mood disorders.

**Table 2 nutrients-12-01892-t002:** Self-reported diagnoses in group 1 (G1) and group 2 (G2).

Self-Reported Diagnosis	Group 1 (*n* = 102)	Group 2 (*n* = 1101)	*p*-Value *
Wheat allergy, *n* (%)	4 (3.9)	0 (0)	-
Dermatitis herpetiformis, *n* (%)	1 (1)	0 (0)	-
Celiac disease, *n* (%)	4 (3.9)	0 (0)	-
Non-celiac gluten/wheat sensitivity, *n* (%)	6 (5.9)	0 (0)	-
Irritable bowel syndrome, *n* (%)	42 (41.1)	189 (17.1)	0.0001
Lactose intolerance, *n* (%)	33 (32.3)	170 (15.4)	0.0001
Chronic diarrhea, *n* (%)	13 (12.7)	37 (3.3)	0.0001
Gastrointestinal cancer, *n* (%)	0 (0)	3 (0.2)	0.990
Allergy, *n* (%)	16 (15.6)	80 (7.2)	0.003
Autoimmune diseases ^†^, *n* (%)	10 (9.8)	49 (4.4)	0.017
Eating disorder ^††^, *n* (%)	11 (10.7)	38 (3.4)	0.0001
Mood disorder ^†††^, *n* (%)	26 (25.4)	102 (9.2)	0.0001

Data are shown as frequency (percentage). * *p*-value obtained by comparison of groups using independent *t*-test. Group 1 included participants reporting symptoms after eating gluten-containing foods; group 2 included participants reporting no discomfort after gluten consumption. ^†^ Includes thyroiditis, diabetes mellitus 1 and psoriasis; ^††^ Includes bulimia and anorexia; ^†††^: Includes depression, anxiety and panic attacks.

**Table 3 nutrients-12-01892-t003:** Self-report of who diagnosed the condition and prescribed a gluten-free diet.

Diagnosis	Diagnosed by (*n*)	Gluten-Free Diet Indicated by (*n*)
Wheat allergy (*n* = 4)	Internist (1) General practitioner (1) Gastroenterologist (2)	General practitioner (1) Self-prescription (1) Nutritionist (2)
Dermatitis herpetiformis (*n* = 1)	Dermatologist (1)	Self-prescription (1)
Celiac disease (*n* = 4)	Gastroenterologist (4)	General practitioner (1) Gastroenterologist (2) Self-prescription (1)
NCG/WS (*n* = 6)	General practitioner (1) Gastroenterologist (4) Bio-holistic Person (1)	Nutritionist (1) Self-prescription (5)

NCG/WS: non-celiac gluten/wheat sensitivity.

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
