# Peer review of "Living with Gluten and Other Food Intolerances: Self-Reported Diagnoses and Management"

_nutrients, 2020, doi:10.3390/nu12061892_

Round 1

Reviewer 1 Report

the paper is very interesting and it gives new informations about the use and abuse of gluten free diet underlying the opportunity to consider also FODMAP diet in selected cases.

I note that a period at page 3 "in cases of wheat allergy (table 2)........" is produced again at page 4

Author Response

Reviwer 1

Thanks for your comment. We aplogogize for the mistake. The sentences were deleted

Reviewer 2 Report

General comments:

This study investigates the prevalence and characteristics of people living with food-related symptoms or disorders in a real-life context. The study is based on an interview and validated survey with self-reported information about medical history. In general, the objectives of this study are interesting and relevant, as gluten-free diet and other restriction diets have gained remarkable popularity during recent years. The results are also interesting, yet the methodology and presentation and interpretation of the results seem to need further revision before publication. In addition, even though I am not a native English speaker and also make many mistakes myself, it seems to me that the English here needs some further editing.

Item-by item comments:

Introduction

Lines 32-35: In my opinion, the story gets started a bit slowly. I think you can move to your main focus a bit faster.

Lines 40-41: Could you refer here how usual it is to follow a GFD without diagnosed celiac disease, either in your own area or e.g. in the US? It would help reader to understand the magnitude of the phenomenon.

Lines 44-45: I’d be careful with bundling celiac disease and NCGS in a same category. NCGS is a disorder with no defined biological basis and in many studies, it has been unclear whether gluten even is the source of symptoms in these patients or is it FODMAPs or some other factor. There is also no evidence that consuming gluten would cause a health hazard to these patients, even though constant symptoms should not be underrated. Throughout the article, I would rather speak celiac disease (and allergies) as its own category as a disease with well-established pathophysiology and separate it from NCGS/NCWS and functional disorders like IBS in which the pathological mechanisms and relations to gluten are much less known.      

Methods

Line 55: Clarify, what means “apparently healthy” in this context? Later you say that participants reported to be diagnosed with disorders, so did you have some exclusion criteria?

Lines 55-58: To me it remains unclear whether the interview happened immediately after recruiting in these public sites (malls etc), or if the interview was arranged later in a more private manner? If it was done immediately, is there a risk that part of the participants declined to attend because they didn’t want to talk about their personal health information in a public place (this could be discussed for example in the discussion)?

Lines 67-70: Dividing to gastrointestinal and extraintestinal condition seems a bit unclear to me. In which category for example celiac disease and IBD belong? Are they gastrointestinal or autoimmune diseases (extraintestinal)? How about allergy, was it extraintestinal irrespective the symptoms that it caused? The same happens in the Table 1. First you say that gastrointestinal/extraintestinal categories refers to “self-reported diagnosis”, but in the foodnote the category includes e.g. chronic diarrhea, which is not a diagnosis, it’s a symptom that is caused by some disorder or is functional if no reason is found. Also, with the extraintestinal category you define it in the foodnote as “reporting one or more extraintestinal symptom”, but allergies and autoimmune diseases are not symptoms, they are diseases with defined pathology and diagnostic criteria. I understand that in this kind of real-life context the diagnoses cannot be confirmed from medical records, but it is nevertheless important that you are consistent whether the participant is reporting symptoms or disorder diagnosed by health care professional. Please clarify this matter.

Results

Lines 104-11 and 11-130: It seems that same paragraph is repeated twice.

Lines 124-135: I think it is useful information who has given the diagnosis/ advised GFD, but could they be presented e.g. as a table? That’d make the results easier to read.  

Lines 165-168: I don’t fully understand the conclusion of the reported reasons to follow GFD. For example celiac disease can cause just minor symptoms. Even though this group by definition was formed of participants reporting symptoms after gluten consumption, is it possible that part of the participants feel that the symptoms are not that bad and they would continue to use gluten if it was not against their health? This kind of discussion of course belongs rather to the Discussion paragraph. than in the Results.

Discussion

Lines 178-179: I don’t understand the meaning of the sentence “Extraintestinal diagnoses related to gluten/wheat consumption were more than twice (44.1%) the ones reported in G2 (21.3%)”. If the G2 was defined as not reporting symptoms after gluten consumption, how could they report extraintestinal diagnoses related to gluten?

Lines 179-187: The whole discussion about neurological symptoms comes as a surprise, as you don’t mention it in results. If you want to discuss them, it’d be better to report the percentages and p-values in the Results section and then interpret the results here in the Discussion. I would also be careful regarding the interpretation of your results here: the neurological symptoms you report in Figure 2 are confusion, headache and tingling sensation. The manifestation that you mention from the literature are ataxia, epilepsy and neuropathy, all being much more objective findings than the symptoms you report here. At least in my clinical opinion, symptoms like headache and numbness/tingling are very common in the general population and very unspecific symptoms. Of course, this does not mean that they’re not worth of further investigations.

Line 192: A spelling error in FODMAPs.

Lines 204-206 and 218-229: Could you give a bit clearer context here, what are the recommendations in Argentina/ Hispanic America considering the beginning of dietary treatment after diagnosing gluten-free diet? Is there a recommendation that nutrition therapist should be consulted etc? What is the general knowledge of celiac disease and GFD among physicians? Are diagnostic tests of celiac disease easily available? You reported earlier in the text that only one person was aware of the risk of gluten contamination and that all patients did not seem to understand GFD as treatment, and also that the advice given by professional were quite vague. I think it is concerning and I’d consider relevant that it was briefly discussed whether this is a common problem and if the diagnostics and dietary advising of celiac disease need further attention in your area.  

Discussion, other comments: You don’t analyze the strengths and limitations of your study. I’d wish a concise discussion especially about your study participants. What strengths and limitations there exists in this kind of real-life context? You also don’t mention in the methods, how easy/hard it was to recruit participants? How big proportion declined and thus, how big the risk of confounding factors in the data is? Is it e.g. possible that people experiencing food intolerance feel the subject more important and hence were more willing to participate?

Figures and tables:

Figure 1: Dermatitis herpetiformis is shortened HD in the picture, even though in footnote is is DH as it should.

Table 1: Please see my comment to gastrointestinal and extraintestinal diagnosis in the Methods section.
